# Further Midpoint Inequalities via Generalized Fractional Operators in Riemann–Liouville Sense

**Abd-Allah Hyder** [1,2,*] , **Hüseyin Budak** [3] and **Areej A. Almoneef** [4]

1   Department of Mathematics, College of Science, King Khalid University, P.O. Box 9004,
    Abha 61413, Saudi Arabia
2   Department of Engineering Mathematics and Physics, Faculty of Engineering, Al-Azhar University,
    Cairo 71524, Egypt
3   Department of Mathematics, Faculty of Science and Arts, Duzce University, Duzce 81620, Turkey
4   Department of Mathematical Sciences, College of Science, Princess Nourah bint Abdulrahman University,
    P.O. Box 84428, Riyadh 11671, Saudi Arabia
*   Correspondence: abahahmed@kku.edu.sa

**Abstract:** In this study, new midpoint-type inequalities are given through recently generalized Riemann–Liouville fractional integrals. Foremost, we present an identity for a class of differentiable functions including the proposed fractional integrals. Then, several midpoint-type inequalities containing generalized Riemann–Liouville fractional integrals are proved by employing the features of convex and concave functions. Furthermore, all obtained results in this study can be compared to previously published results.

**Keywords:** generalized fractional operators; midpoint inequalities; Hermite–Hadamard inequality

## 1. Introduction

Fractional calculus is an area of mathematics that expands the traditional derivative and integral ideas to noninteger orders. In recent decades, it has piqued the curiosity of mathematicians, physicists, and engineers [1–3]. In a fluid-dynamic traffic model, fractional derivatives can be utilized to simulate the irregular oscillation of earthquakes and to compensate for the inadequacies induced by the assumption of a continuous traffic flow. Fractional derivatives are also used to model a wide range of chemical processes, as well as mathematical biology and other physics and engineering problems [4–8]. Further, it has been demonstrated that several fractional systems produce results that are more appropriate than those produced by corresponding systems having integer derivatives [9,10].

New studies have concentrated on developing a class of fractional integral operators and their applicability in a variety of scientific disciplines. Using only the derivative's fundamental limit formulation, a newly well-behaved straightforward fractional derivative known as the conformable derivative was developed in [11]. Some significant requirements that cannot be fulfilled by the Riemann–Liouville and Caputo definitions are fulfilled by the conformable derivative. Nevertheless, in [12], the author demonstrated that the conformable approach in [11] could not yield good results when compared to the Caputo definition for specific functions. This flaw in the conformable definition was avoided by some extensions of the conformable approach [13,14]. In addition, employing exponential and Mittag–Leffler functions in the kernels, several scholars created novel expanded fractional operators [15–19].

The Hermite–Hadamard inequality, which is the initial conclusion of convex functions with a straightforward geometric explanation and different applications, has recently attracted considerable interest in both elementary and advanced mathematics. The Hermite–

Hadamard inequality declares that if $\phi : Z \subset \mathbb{R} \to \mathbb{R}$ is a convex mapping and $\epsilon_1, \epsilon_2 \in Z$ with $\epsilon_1 < \epsilon_2$, then

$$\phi\left(\frac{\epsilon_1 + \epsilon_2}{2}\right) \leq \frac{1}{\epsilon_2 - \epsilon_1} \int_{\epsilon_1}^{\epsilon_2} \phi(\varsigma) d\varsigma \leq \frac{\phi(\epsilon_1) + \phi(\epsilon_2)}{2}. \tag{1}$$

If $\phi$ is concave, the two inequalities are valid in the reverse orientation.

Since the discovery of inequality (1), it has been the focus of substantial research, and a number of articles have been published that offer notable expansions, generalizations, and improvements for a new category of convex functions. Please see for some instances [20–26].

Several scientists investigated the Hermite–Hadamard inequality utilizing fractional operators and produced a variety of extensions and enhancements. Sarikaya and Alp [27] used local fractional integrals to investigate the Hermite–Hadamard–Fejér integral inequalities for generic convex mappings. Kwun et al. [28] explored generalized Riemann–Liouville fractional integrals connected with Ostrowski type inequalities and Hadamard error constraints. Budak et al. [29] defined new Riemann–Liouville fractional integrals for interval-valued functions on coordinates. Using these specified fractional integrals, they also established Hermite–Hadamard and other related inequalities for coordinated convex interval-valued functions. Hyder et al. [30] recently used more general fractional operators to demonstrate further fractional inequalities in the Hermite–Hadamard and Minkowski contexts. For some more results, one can refer to [31–33].

Following are some concepts and foundations of fractional calculus that are utilized later in this research.

**Definition 1** ([17]). *Let $\phi \in L^1[\varpi, \varrho]$, $\varpi, \varrho \in \mathbb{R}$ with $\varpi < \varrho$. The Riemann–Liouville integrals $J_{\varpi+}^{\eta}\phi$ and $J_{\varrho-}^{\eta}\phi$ of order $\eta > 0$ are defined by*

$$J_{\varpi+}^{\eta}\phi(\xi) = \frac{1}{\Gamma(\eta)} \int_{\varpi}^{\xi} (\xi - \varsigma)^{\eta-1} \phi(\varsigma) d\varsigma, \quad \xi > \varpi, \tag{2}$$

*and*

$$J_{\varrho-}^{\eta}\phi(\xi) = \frac{1}{\Gamma(\eta)} \int_{\xi}^{\varrho} (\varsigma - \xi)^{\eta-1} \phi(\varsigma) d\varsigma, \quad \xi < \varrho, \tag{3}$$

*respectively. Here, $\Gamma$ denotes the gamma function and $J_{\varpi+}^{0}\phi(\xi) = J_{\varrho-}^{0}\phi(\xi) = \phi(\xi)$.*

Jarad et al. [18] introduced the following generalized fractional integral operators. They also provided certain characteristics and relationships between these operators and several other fractional operators in the literature

**Definition 2** ([18]). *Let $\eta \in \mathbb{C}$, $Re(\eta) > 0$, and $\theta \in (0,1]$. For $\phi \in L^1[\varpi, \varrho]$, the generalized fractional Riemann–Liouville integrals ${}_{\varpi}^{\eta}Y^{\theta}\phi$ and ${}^{\eta}Y_{\varrho}^{\theta}\phi$, of order $(\eta, \theta)$, are defined by*

$${}_{\varpi}^{\eta}Y^{\theta}\phi(\xi) = \frac{1}{\Gamma(\eta)} \int_{\varpi}^{\xi} \left(\frac{(\xi - \varpi)^{\theta} - (\varsigma - \varpi)^{\theta}}{\theta}\right)^{\eta-1} \frac{\phi(\varsigma)}{(\varsigma - \varpi)^{1-\theta}} d\varsigma, \quad \xi > \varpi, \tag{4}$$

*and*

$${}^{\eta}Y_{\varrho}^{\theta}\phi(\xi) = \frac{1}{\Gamma(\eta)} \int_{\xi}^{\varrho} \left(\frac{(\varrho - \xi)^{\theta} - (\varrho - \varsigma)^{\theta}}{\theta}\right)^{\eta-1} \frac{\phi(\varsigma)}{(\varrho - \varsigma)^{1-\theta}} d\varsigma, \quad \xi < \varrho, \tag{5}$$

*respectively.*

**Remark 1** ([18]). *When $\theta = 1$, $\varpi = 0$, the fractional operator in (4) and the Riemann–Liouville integral in (2) are the same. Furthermore, if $\theta = 1$, $\varrho = 0$, the fractional operator in (5) reduces to the Riemann–Liouville integral in (3).*

Using the fractional integrals in (4) and (5), Set et al. [34] presented a notable Hermite–Hadamard integral inequality as below:

**Theorem 1** ([34]). *Assume $\phi$ is a positive convex function from $[\varpi, \varrho]$ into $\mathbb{R}$. If $0 \leq \varpi < \varrho$ and $\phi \in L^1[\varpi, \varrho]$, then the next inequality holds for the generalized fractional integrals ${}^{\eta}_{\varpi}Y^{\theta}$ and ${}^{\eta}Y^{\theta}_{\varrho}$:*

$$\phi\left(\frac{\varpi + \varrho}{2}\right) \leq \frac{\Gamma(\eta+1)\theta^{\eta}}{2(\varrho-\varpi)^{\theta\eta}}\left[{}^{\eta}_{\varpi}Y^{\theta}\phi(\varrho) + {}^{\eta}Y^{\theta}_{\varrho}\phi(\varpi)\right] \leq \frac{\phi(\varpi) + \phi(\varrho)}{2}, \qquad (6)$$

*where $Re(\eta) > 0$ and $\theta \in (0, 1]$.*

Furthermore, the Hermite–Hadamard inequality of a positive convex function that involves the fractional operators (4) and (5) was represented by Gözpınar as follows:

**Theorem 2** ([35]). *Suppose $\phi : [\varpi, \varrho] \to \mathbb{R}$ is a positive convex function with $0 \leq \varpi < \varrho$ and $\phi \in L^1[\varpi, \varrho]$. If $Re(\eta) > 0$ and $\theta \in (0, 1]$, then we get the inequality:*

$$\phi\left(\frac{\varpi + \varrho}{2}\right) \leq \frac{2^{\theta\eta-1}\Gamma(\eta+1)\theta^{\eta}}{(\varrho-\varpi)^{\theta\eta}}\left[{}^{\eta}_{\frac{\varpi+\varrho}{2}}Y^{\theta}\phi(\varrho) + {}^{\eta}Y^{\theta}_{\frac{\varpi+\varrho}{2}}\phi(\varpi)\right] \leq \frac{\phi(\varpi) + \phi(\varrho)}{2}. \qquad (7)$$

In the current study, we present new midpoint inequalities through the generalized Riemann–Liouville fractional integrals (4) and (5). For a class of differentiable functions, we create a new identity including the proposed fractional integrals. Hence, by employing convex and concave mappings, several generalized midpoint inequalities are obtained. Furthermore, our results can be compared to previously known results.

This paper is constructed as follows: In Section 2, we present the main results. Precisely, we create a new identity concerning a class of differentiable functions and involving the suggested fractional integrals. Consequently, by utilizing convex and concave mappings, diverse generalized midpoint inequalities are obtained. Section 3 involves some conclusions.

## 2. Main Results

We start with proving the next Lemma which is utilized frequently throughout this section.

**Lemma 1.** *Let $\phi$ be a function from $[\varpi, \varrho]$ into $\mathbb{R}$ with $\varpi < \varrho$. If $\phi \in L^1[\varpi, \varrho]$ and differentiable on $(\varpi, \varrho)$, then the next identity holds for each $\xi \in [\varpi, \varrho]$:*

$$\begin{aligned}
&\frac{\theta^{\eta}\Gamma(\eta+1)}{\varrho-\varpi}\left[(\xi-\varpi)^{1-\theta\eta}\,{}^{\eta}Y^{\theta}_{\varrho}\phi(\varpi+\varrho-\xi) + (\varpi-\xi)^{1-\eta\theta}\,{}^{\eta}_{\varpi}Y^{\theta}\phi(\varpi+\varrho-\xi)\right] - \phi(\varpi+\varrho-\xi) \\
&= \frac{\theta^{\eta}(\xi-\varpi)^2}{\varrho-\varpi}\int_0^1\left[\frac{1}{\theta^{\eta}} - \left(\frac{1-(1-\varsigma)^{\theta}}{\theta}\right)^{\eta}\right]\phi'(\varsigma\varrho + (1-\varsigma)(\varpi+\varrho-\xi))d\varsigma \\
&\quad + \frac{\theta^{\eta}(\varpi-\xi)^2}{\varrho-\varpi}\int_0^1\left[\left(\frac{1-(1-\varsigma)^{\theta}}{\theta}\right)^{\eta} - \frac{1}{\theta^{\eta}}\right]\phi'(\varsigma\varpi + (1-\varsigma)(\varpi+\varrho-\xi))d\varsigma.
\end{aligned} \qquad (8)$$

**Proof.** Applying the integration by parts, we get

$$\int\limits_0^1 \left[ \frac{1}{\theta^\eta} - \left( \frac{1 - (1 - \varsigma)^\theta}{\theta} \right)^\eta \right] \phi'(\varsigma\varrho + (1 - \varsigma)(\varpi + \varrho - \xi)) d\varsigma$$

$$= \left[ \frac{1}{\theta^\eta} - \left( \frac{1 - (1 - \varsigma)^\theta}{\theta} \right)^\eta \right] \frac{\phi(\varsigma\varrho + (1 - \varsigma)(\varpi + \varrho - \xi))}{\xi - \varpi} \Bigg|_0^1$$

$$+ \frac{\eta}{\xi - \varpi} \int\limits_0^1 \left( \frac{1 - (1 - \varsigma)^\theta}{\theta} \right)^{\eta - 1} (1 - \varsigma)^{\theta - 1} \phi(\varsigma\varrho + (1 - \varsigma)(\varpi + \varrho - \xi)) d\varsigma \tag{9}$$

$$= -\frac{\phi(\varpi + \varrho - \xi)}{\theta^\eta (\xi - \varpi)} + \frac{\eta}{(\xi - \varpi)^{\theta\eta + 1}} \int\limits_{\varpi + \varrho - \xi}^{\varrho} \left( \frac{(\varrho - \xi)^\theta - (\varrho - \varsigma)^\theta}{\theta} \right)^{\eta - 1} \phi(\varsigma) \frac{d\varsigma}{(\varpi - \varsigma)^{1 - \theta}}$$

$$= -\frac{\phi(\varpi + \varrho - \xi)}{\theta^\eta (\xi - \varpi)} + \frac{\Gamma(\eta + 1)}{(\xi - \varpi)^{\theta\eta + 1}} \, {}^\eta Y_\varrho^\theta \phi(\varpi + \varrho - \xi).$$

Likewise, we have

$$\int\limits_0^1 \left[ \left( \frac{1 - (1 - \varsigma)^\theta}{\theta} \right)^\eta - \frac{1}{\theta^\eta} \right] \phi'(\varsigma\varpi + (1 - \varsigma)(\varpi + \varrho - \xi)) d\varsigma \tag{10}$$

$$= -\frac{\phi(\varpi + \varrho - \xi)}{\theta^\eta (\varpi - \xi)} + \frac{\Gamma(1 + \theta)}{(\varpi - \xi)^{\eta\theta + 1}} \, {}^\eta_a Y^\theta \phi(\varpi + \varrho - \xi).$$

By Equalities (9) and (10), the required identity (8) is obtained. □

**Theorem 3.** *Let $\phi$ be a function from $[\varpi, \varrho]$ into $\mathbb{R}$. If $\phi$ is differentiable on $(\varpi, \varrho)$ and $|\phi'|$ is convex on $[\varpi, \varrho]$, then the next inequality holds for the fractional integrals $\, {}^\eta_\varpi Y^\theta$, $\, {}^\eta Y_\varrho^\theta$, and $\xi \in [\varpi, \varrho]$:*

$$\left| \frac{\theta^\eta \Gamma(\eta + 1)}{\varrho - \varpi} \left[ (\xi - \varpi)^{1 - \theta\eta} \, {}^\eta Y_\varrho^\theta \phi(\varpi + \varrho - \xi) + (\varpi - \xi)^{1 - \eta\theta} \, {}^\eta_\varpi Y^\theta \phi(\varpi + \varrho - \xi) \right] - \phi(\varpi + \varrho - \xi) \right|$$

$$\leq \frac{(\xi - \varpi)^2}{\varrho - \varpi} \left\{ \left[ \frac{1}{2} - \frac{1}{\theta} \left( B\left( \eta + 1, \frac{1}{\theta} \right) - B\left( \eta + 1, \frac{2}{\theta} \right) \right) \right] |\phi'(\varrho)| \right.$$

$$\left. + \left[ \frac{1}{2} - \frac{1}{\theta} B\left( \eta + 1, \frac{2}{\theta} \right) \right] |\phi'(\varpi + \varrho - \xi)| \right\} \tag{11}$$

$$+ \frac{\theta^\eta (\varpi - \xi)^2}{\varrho - \varpi} \left\{ \left[ \frac{1}{2} - \frac{1}{\theta} \left( B\left( \eta + 1, \frac{1}{\theta} \right) - B\left( \eta + 1, \frac{2}{\theta} \right) \right) \right] |\phi'(\varpi)| \right.$$

$$\left. + \left[ \frac{1}{2} - \frac{1}{\theta} B\left( \eta + 1, \frac{2}{\theta} \right) \right] |\phi'(\varpi + \varrho - \xi)| \right\},$$

*where $B(z_1, z_2) = \int_0^1 \varsigma^{z_1 - 1} (1 - \varsigma)^{z_2 - 1} d\varsigma$ is the Euler Beta function.*

**Proof.** According to Lemma 1, we get

$$
\left| \frac{\theta^{\eta} \Gamma(\eta+1)}{\varrho - \varpi} \left[ (\xi - \varpi)^{1-\theta\eta} \, {}^{\eta}Y_{\varrho}^{\theta}\phi(\varpi + \varrho - \xi) + (\varpi - \xi)^{1-\eta\theta} \, {}^{\eta}_{\varpi}Y^{\theta}\phi(\varpi + \varrho - \xi) \right] - \phi(\varpi + \varrho - \xi) \right| \tag{12}
$$

$$
\leq \quad \frac{\theta^{\eta}(\xi - \varpi)^2}{\varrho - \varpi} \int_0^1 \left| \frac{1}{\theta^{\eta}} - \left( \frac{1 - (1 - \varsigma)^{\theta}}{\theta} \right)^{\eta} \right| |\phi'(\varsigma\varrho + (1 - \varsigma)(\varpi + \varrho - \xi))| d\varsigma
$$

$$
+ \frac{\theta^{\eta}(\varpi - \xi)^2}{\varrho - \varpi} \int_0^1 \left| \left( \frac{1 - (1 - \varsigma)^{\theta}}{\theta} \right)^{\eta} - \frac{1}{\theta^{\eta}} \right| |\phi'(\varsigma\varpi + (1 - \varsigma)(\varpi + \varrho - \xi))| d\varsigma.
$$

From the convexity of $|\phi'|$, we obtain

$$
\int_0^1 \left| \frac{1}{\theta^{\eta}} - \left( \frac{1 - (1 - \varsigma)^{\theta}}{\theta} \right)^{\eta} \right| |\phi'(\varsigma\varrho + (1 - \varsigma)(\varpi + \varrho - \xi))| d\varsigma
$$

$$
\leq \quad \frac{1}{\theta^{\eta}} \int_0^1 \left[ 1 - \left( 1 - (1 - \varsigma)^{\theta} \right)^{\eta} \right] \left[ \varsigma|\phi'(\varrho)| + (1 - \varsigma)|\phi'(\varpi + \varrho - \xi)| \right] d\varsigma \tag{13}
$$

$$
= \quad \frac{1}{\theta^{\eta}} \left\{ \left[ \frac{1}{2} - \frac{1}{\theta} \left( B\left( \eta + 1, \frac{1}{\theta} \right) - B\left( \eta + 1, \frac{2}{\theta} \right) \right) \right] |\phi'(\varrho)| \right.
$$

$$
\left. + \left[ \frac{1}{2} - \frac{1}{\theta} B\left( \eta + 1, \frac{2}{\theta} \right) \right] |\phi'(\varpi + \varrho - \xi)| \right\}.
$$

Similarly, we have

$$
\int_0^1 \left| \left( \frac{1 - (1 - \varsigma)^{\theta}}{\theta} \right)^{\eta} - \frac{1}{\theta^{\eta}} \right| |\phi'(\varsigma\varpi + (1 - \varsigma)(\varpi + \varrho - \xi))| d\varsigma
$$

$$
\leq \quad \frac{1}{\theta^{\eta}} \left\{ \left[ \frac{1}{2} - \frac{1}{\theta} \left( B\left( \eta + 1, \frac{1}{\theta} \right) - B\left( \eta + 1, \frac{2}{\theta} \right) \right) \right] |\phi'(\varpi)| \right. \tag{14}
$$

$$
\left. + \left[ \frac{1}{2} - \frac{1}{\theta} B\left( \eta + 1, \frac{2}{\theta} \right) \right] |\phi'(\varpi + \varrho - \xi)| \right\}.
$$

By inserting inequalities (13) and (14) in (12), the desired inequality (11) is obtained. □

**Corollary 1.** *If we choose $\theta = 1$ in Theorem 3, then we have the following inequality for Riemann–Liouville fractional integrals*

$$
\left| \frac{\Gamma(\eta+1)}{\varrho - \varpi} \left[ (\xi - \varpi)^{1-\eta} \, J_{\varrho-}^{\eta}\phi(\varpi + \varrho - \xi) + (\varpi - \xi)^{1-\eta} \, J_{\varpi+}^{\eta}\phi(\varpi + \varrho - \xi) \right] - \phi(\varpi + \varrho - \xi) \right|
$$

$$
\leq \quad \frac{\eta}{2(\varrho - \varpi)(\eta+2)} \left( (\xi - \varpi)^2 |\phi'(\varrho)| + (\varpi - \xi)^2 |\phi'(\varpi)| \right) \tag{15}
$$

$$
+ \left( \frac{1}{2} - \frac{1}{(\eta+1)(\eta+2)} \right) \left( \frac{(\xi - \varpi)^2 + (\varpi - \xi)^2}{\varrho - \varpi} \right) |\phi'(\varpi + \varrho - \xi)|.
$$

**Remark 2.** *If we assign $\eta = 1$ in Corollary 1, then Corollary 1 reduces to [36] (Theorem 5, for $q = 1$).*

**Corollary 2.** *Consider the assumptions of Theorem 3. If $\xi = \frac{\omega + \varrho}{2}$, we get the next inequality:*

$$\left| \frac{2^{\theta\eta-1}\Gamma(\eta+1)}{(\varrho-\omega)^{\theta\eta}} \left[ {}^{\eta}Y_{\varrho}^{\theta}\phi\left(\frac{\omega+\varrho}{2}\right) + {}_{\omega}^{\eta}Y^{\theta}\phi\left(\frac{\omega+\varrho}{2}\right) \right] - \phi\left(\frac{\omega+\varrho}{2}\right) \right| \tag{16}$$

$$\leq \frac{\varrho-\omega}{4}\left(1 - \frac{1}{\theta}B\left(\eta+1,\frac{1}{\theta}\right)\right)\left[|\phi'(\varrho)| + |\phi'(\omega)|\right].$$

**Remark 3.** *If we take $\theta = \eta = 1$ in Corollary 2, then Corollary 2 reduces to [37] (Theorem 2.2).*

**Example 1.** *Let $[\omega, \varrho] = [1, 2]$ and let $\xi = \frac{3}{2}$. Consider the function $\phi : [0, 1] \to \mathbb{R}$ defined by $\phi(\varsigma) = \frac{\varsigma^3}{3}$. Then, $\phi'(\varsigma) = \varsigma^2$ and $|\phi'|$ is convex on $[1, 2]$. Under these assumptions,*

$$J_{\varrho-}^{\eta}\phi(\omega+\varrho-\xi) = \frac{1}{\Gamma(\eta)}\int_{\frac{3}{2}}^{2}\left(\varsigma - \frac{3}{2}\right)^{\eta-1}\frac{\varsigma^3}{3}d\varsigma = \frac{32\eta^3 + 168\eta^2 + 244\eta + 81}{3 \cdot 2^{\eta+2}\Gamma(\eta+4)}, \tag{17}$$

*and*

$$J_{\omega+}^{\eta}\phi(\omega+\varrho-\xi) = \frac{1}{\Gamma(\eta)}\int_{1}^{\frac{3}{2}}\left(\frac{3}{2} - \varsigma\right)^{\eta-1}\frac{\varsigma^3}{3}d\varsigma = \frac{4\eta^3 + 30\eta^2 + 80\eta + 81}{3 \cdot 2^{\eta+2}\Gamma(\eta+4)}. \tag{18}$$

*The left-hand side of (15) reduces to*

$$\left| \frac{\Gamma(\eta+1)}{\varrho-\omega}\left[ (\xi-\omega)^{1-\eta} J_{b-}^{\theta}\phi(\omega+\varrho-\xi) + (\omega-\xi)^{1-\eta} J_{a+}^{\theta}\phi(\omega+\varrho-\xi) \right] - \phi(\omega+\varrho-\xi) \right|$$

$$= \left| \Gamma(\eta+1)\left[ \left(\frac{1}{2}\right)^{1-\eta}\frac{32\eta^3 + 168\eta^2 + 244\eta + 81}{3 \cdot 2^{\eta+2}\Gamma(\eta+4)} + \left(\frac{1}{2}\right)^{1-\eta}\frac{4\eta^3 + 30\eta^2 + 80\eta + 81}{3 \cdot 2^{\eta+2}\Gamma(\eta+4)} \right] - \frac{9}{8} \right| \tag{19}$$

$$= \left| \frac{18\eta^3 + 99\eta^2 + 162\eta + 81}{24(\eta+1)(\eta+2)(\eta+3)} - \frac{9}{8} \right|.$$

*Similarly, the right-hand side of (15) reduces to*

$$\left( \frac{\eta}{2(\varrho-\omega)(\eta+2)} \right)\left( (\xi-\omega)^2|\phi'(\varrho)| + (\omega-\xi)^2|\phi'(\omega)| \right)$$

$$+ \left( \frac{1}{2} - \frac{1}{(\eta+1)(\eta+2)} \right)\left( \frac{(\xi-\omega)^2 + (\omega-\xi)^2}{\varrho-\omega} \right)|\phi'(\omega+\varrho-\xi)| \tag{20}$$

$$= \frac{5\eta^2 + 5\eta - 9}{8(\eta+2)} + \frac{9}{16}.$$

*By inequality (15), we have the inequality*

$$\left| \frac{18\eta^3 + 99\eta^2 + 162\eta + 81}{24(\eta+1)(\eta+2)(\eta+3)} - \frac{9}{8} \right| \leq \frac{5\eta^2 + 5\eta - 9}{8(\eta+2)} + \frac{9}{16}. \tag{21}$$

*One can see the validity of inequality (21) in Figure 1.*

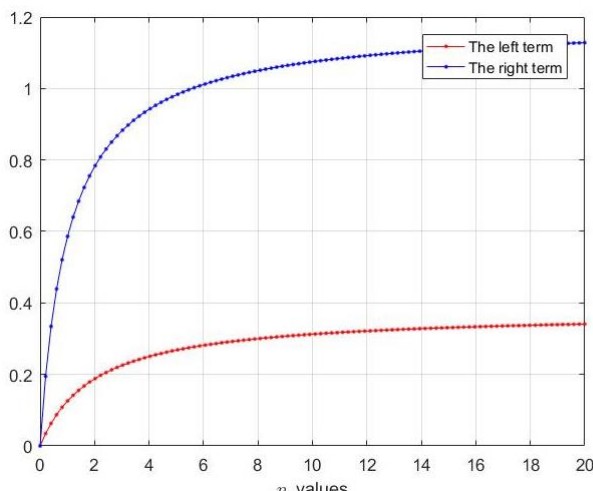

**Figure 1.** An example of inequality (15) depending on $\eta$ , computed and plotted with MATLAB.

**Theorem 4.** *Let $\phi$ be a function from $[\varpi, \varrho]$ into $\mathbb{R}$. If $\phi$ is differentiable on $(\varpi, \varrho)$ and $|\phi'|^q$ is convex for $\xi \in [\varpi, \varrho]$ and $q > 1$, then the next inequality holds for the fractional integrals $_\varpi^\eta Y^\theta$, $^\eta Y_\varrho^\theta$, and $\xi \in [\varpi, \varrho]$:*

$$\left| \frac{\theta^\eta \Gamma(\eta+1)}{\varrho - \varpi} \left[ (\xi - \varpi)^{1-\theta\eta} \, {}^\eta Y_\varrho^\theta \phi(\varpi + \varrho - \xi) + (\varpi - \xi)^{1-\eta\theta} \, {}_\varpi^\eta Y^\theta \phi(\varpi + \varrho - \xi) \right] - \phi(\varpi + \varrho - \xi) \right|$$

$$\leq \frac{(\xi - \varpi)^2}{\varrho - \varpi} \left( 1 - \frac{1}{\theta} B\left( p\eta + 1, \frac{1}{\theta} \right) \right)^{\frac{1}{p}} \left( \frac{|\phi'(\varrho)|^q + |\phi'(\varpi + \varrho - \xi)|^q}{2} \right)^{\frac{1}{q}} \tag{22}$$

$$+ \frac{(\varpi - \xi)^2}{\varrho - \varpi} \left( 1 - \frac{1}{\theta} B\left( p\eta + 1, \frac{1}{\theta} \right) \right)^{\frac{1}{p}} \left( \frac{|\phi'(\varpi)|^q + |\phi'(\varpi + \varrho - \xi)|^q}{2} \right)^{\frac{1}{q}},$$

*where $B(\cdot, \cdot)$ is Euler's beta function and $\frac{1}{q} + \frac{1}{p} = 1$.*

**Proof.** According to Hölder's inequality and the convexity of $|\phi'|^q$, we get

$$\int_0^1 \left| \frac{1}{\theta^\eta} - \left( \frac{1 - (1-\varsigma)^\theta}{\theta} \right)^\eta \right| \left| \phi'(\varsigma\varrho + (1-\varsigma)(\varpi + \varrho - \xi)) \right| d\varsigma$$

$$\leq \left( \int_0^1 \left| \frac{1}{\theta^\eta} - \left( \frac{1 - (1-\varsigma)^\theta}{\theta} \right)^\eta \right|^p d\varsigma \right)^{\frac{1}{p}} \left( \int_0^1 |\phi'(\varsigma\varrho + (1-\varsigma)(\varpi + \varrho - \xi))|^q d\varsigma \right)^{\frac{1}{q}} \tag{23}$$

$$\leq \frac{1}{\theta^\eta} \left( \int_0^1 \left( 1 - \left( 1 - (1-\varsigma)^\theta \right)^{p\eta} \right) d\varsigma \right)^{\frac{1}{p}} \left( \int_0^1 \left[ \varsigma |\phi'(\varrho)|^q + (1-\varsigma) |\phi'(\varpi + \varrho - \xi)|^q \right] d\varsigma \right)^{\frac{1}{q}}$$

$$= \frac{1}{\theta^\eta} \left( 1 - \frac{1}{\theta} B\left( p\eta + 1, \frac{1}{\theta} \right) \right)^{\frac{1}{q}} \left( \frac{|\phi'(\varrho)|^q + |\phi'(\varpi + \varrho - \xi)|^q}{2} \right)^{\frac{1}{q}}.$$

Here, we utilize the fact that

$$(m - n)^k \leq m^k - n^k, \tag{24}$$

for any $m > n \geq 0$ and $k \geq 1$.

Likewise, we have

$$\int\limits_0^1 \left| \left( \frac{1 - (1 - \varsigma)^\theta}{\theta} \right)^\eta - \frac{1}{\theta^\eta} \right| |\phi'(\varsigma\omega + (1 - \varsigma)(\omega + \varrho - \xi))| d\varsigma \tag{25}$$

$$\leq \frac{1}{\theta^\eta} \left( 1 - \frac{1}{\theta} B\left( p\eta + 1, \frac{1}{\theta} \right) \right)^{\frac{1}{p}} \left( \frac{|\phi'(\omega)|^q + |\phi'(\omega + \varrho - \xi)|^q}{2} \right)^{\frac{1}{q}}.$$

Substituting inequalities (23) and (25) in (12), we get the required inequality (22). □

**Remark 4.** *If we choose $\theta = 1$ in Theorem 4, then Theorem 4 reduces to [38] (Theorem 3).*

**Corollary 3.** *In view of the assumptions of Theorem 4. If $\xi = \frac{\omega + \varrho}{2}$, we gain the inequality below*

$$\left| \frac{2^{\theta\eta - 1}\Gamma(\theta + 1)}{(\varrho - \omega)^{\theta\eta}} \left[ {}^\eta Y_\varrho^\theta \phi\left( \frac{\omega + \varrho}{2} \right) + {}_\omega^\eta Y^\theta \phi\left( \frac{\omega + \varrho}{2} \right) \right] - \phi\left( \frac{\omega + \varrho}{2} \right) \right|$$

$$\leq \frac{\varrho - \omega}{4} \left( 1 - \frac{1}{\theta} B\left( p\eta + 1, \frac{1}{\theta} \right) \right)^{\frac{1}{p}} \left[ \left( \frac{3|\phi'(\omega)|^q + |\phi'(\varrho)|^q}{4} \right)^{\frac{1}{q}} + \left( \frac{3|\phi'(\varrho)|^q + |\phi'(\omega)|^q}{4} \right)^{\frac{1}{q}} \right] \tag{26}$$

$$\leq \frac{\varrho - \omega}{4} \left( 4 - \frac{4}{\theta} B\left( p\eta + 1, \frac{1}{\theta} \right) \right)^{\frac{1}{p}} [|\phi'(\omega)| + |\phi'(\varrho)|].$$

**Proof.** It is obvious that the first inequality in (26) can be acquired from the convexity of $|\phi'|^q$. The second inequality can be obtained directly by letting $\omega_1 = 3|\phi'(\omega)|^q$, $\varrho_1 = |\phi'(\varrho)|^q$, $\omega_2 = |\phi'(\omega)|^q$, and $\varrho_2 = 3|\phi'(\varrho)|^q$ and applying the inequality:

$$\sum_{k=1}^n (\omega_k + \varrho_k)^s \leq \sum_{k=1}^n \omega_k^s + \sum_{k=1}^n \varrho_k^s, \quad 0 \leq s < 1. \tag{27}$$

□

**Remark 5.** *If we choose $\theta = 1$ in Corollary 3, then Corollary 3 reduces to [38] (Corollary 1).*

**Theorem 5.** *Assume $\phi$ is a function from $[\omega, \varrho]$ into $\mathbb{R}$. If $\phi$ is differentiable on $(\omega, \varrho)$ and $|\phi'|^q$ is convex for $\xi \in [\omega, \varrho]$ and for some $q > 1$, then the next inequality holds for the fractional integrals ${}_\omega^\eta Y^\theta$, ${}^\eta Y_\varrho^\theta$, and $\xi \in [\omega, \varrho]$:*

$$\left| \frac{\theta^\eta \Gamma(\eta + 1)}{\varrho - \omega} \left[ (\xi - \omega)^{1 - \theta\eta} \, {}^\eta Y_\varrho^\theta \phi(\omega + \varrho - \xi) + (\omega - \xi)^{1 - \eta\theta} \, {}_\omega^\eta Y^\theta \phi(\omega + \varrho - \xi) \right] - \phi(\omega + \varrho - \xi) \right|$$

$$\leq \frac{1}{\varrho - \omega} \left( 1 - \frac{1}{\theta} B\left( \eta + 1, \frac{1}{\theta} \right) \right)^{1 - \frac{1}{q}}$$

$$\times \left[ (\xi - \omega)^2 \left\{ \left[ \frac{1}{2} - \frac{1}{\theta} \left( B\left( \eta + 1, \frac{1}{\theta} \right) - B\left( \eta + 1, \frac{2}{\theta} \right) \right) \right] |\phi'(\varrho)|^q \right. \right. \tag{28}$$

$$\left. + \left[ \frac{1}{2} - \frac{1}{\theta} B\left( \eta + 1, \frac{2}{\theta} \right) \right] |\phi'(\omega + \varrho - \xi)|^q \right\}^{\frac{1}{q}}$$

$$+ (\omega - \xi)^2 \left\{ \left[ \frac{1}{2} - \frac{1}{\theta} \left( B\left( \eta + 1, \frac{1}{\theta} \right) - B\left( \eta + 1, \frac{2}{\theta} \right) \right) \right] |\phi'(\omega)|^q \right.$$

$$\left. \left. + \left[ \frac{1}{2} - \frac{1}{\theta} B\left( \eta + 1, \frac{2}{\theta} \right) \right] |\phi'(\omega + \varrho - \xi)|^q \right\}^{\frac{1}{q}} \right].$$

**Proof.** Using the power mean inequality in inequality (12), we get

$$\left| \frac{\theta^\eta \Gamma(\eta+1)}{\varrho - \varpi} \left[ (\xi - \varpi)^{1-\theta\eta} \, {}^\eta Y_\varrho^\theta \phi(\varpi + \varrho - \xi) + (\varpi - \xi)^{1-\eta\theta} \, {}^\eta_\varpi Y^\theta \phi(\varpi + \varrho - \xi) \right] - \phi(\varpi + \varrho - \xi) \right|$$

$$\leq \frac{\theta^\eta (\xi - \varpi)^2}{\varrho - \varpi} \left( \int_0^1 \left| \frac{1}{\theta^\eta} - \left( \frac{1 - (1-\varsigma)^\theta}{\theta} \right)^\eta \right| d\varsigma \right)^{1 - \frac{1}{q}}$$

$$\times \left( \int_0^1 \left| \frac{1}{\theta^\eta} - \left( \frac{1 - (1-\varsigma)^\theta}{\theta} \right)^\eta \right| \left| \phi'(\varsigma\varrho + (1-\varsigma)(\varpi + \varrho - \xi)) \right|^q d\varsigma \right)^{\frac{1}{q}} \tag{29}$$

$$+ \frac{\theta^\eta (\varpi - \xi)^2}{\varrho - \varpi} \left( \int_0^1 \left| \frac{1}{\theta^\eta} - \left( \frac{1 - (1-\varsigma)^\theta}{\theta} \right)^\eta \right| d\varsigma \right)^{1 - \frac{1}{q}}$$

$$\times \left( \int_0^1 \left| \frac{1}{\theta^\eta} - \left( \frac{1 - (1-\varsigma)^\theta}{\theta} \right)^\eta \right| \left| \phi'(\varsigma\varpi + (1-\varsigma)(\varpi + \varrho - \xi)) \right|^q d\varsigma \right)^{\frac{1}{q}}.$$

By the convexity of $|\phi'|^q$, we have

$$\int_0^1 \left| \frac{1}{\theta^\eta} - \left( \frac{1 - (1-\varsigma)^\theta}{\theta} \right)^\eta \right| \left| \phi'(\varsigma\varrho + (1-\varsigma)(\varpi + \varrho - \xi)) \right|^q d\varsigma$$

$$\leq \int_0^1 \left( \frac{1}{\theta^\eta} - \left( \frac{1 - (1-\varsigma)^\theta}{\theta} \right)^\eta \right) \left[ \varsigma |\phi'(\varrho)|^q + (1-\varsigma)|\phi'(\varpi + \varrho - \xi)|^q \right] d\varsigma$$

$$= \frac{1}{\theta^\eta} \left\{ \left[ \frac{1}{2} - \frac{1}{\theta} \left( B\left( \eta + 1, \frac{1}{\theta} \right) - B\left( \eta + 1, \frac{2}{\theta} \right) \right) \right] |\phi'(\varrho)|^q \right. \tag{30}$$

$$\left. + \left[ \frac{1}{2} - \frac{1}{\theta} B\left( \eta + 1, \frac{2}{\theta} \right) \right] |\phi'(\varpi + \varrho - \xi)|^q \right\}.$$

Similarly, we obtain

$$\int_0^1 \left| \frac{1}{\theta^\eta} - \left( \frac{1 - (1-\varsigma)^\theta}{\theta} \right)^\eta \right| \left| \phi'(\varsigma\varpi + (1-\varsigma)(\varpi + \varrho - \xi)) \right|^q d\varsigma$$

$$\leq \frac{1}{\theta^\eta} \left\{ \left[ \frac{1}{2} - \frac{1}{\theta} \left( B\left( \eta + 1, \frac{1}{\theta} \right) - B\left( \eta + 1, \frac{2}{\theta} \right) \right) \right] |\phi'(\varpi)|^q \right. \tag{31}$$

$$\left. + \left[ \frac{1}{2} - \frac{1}{\theta} B\left( \eta + 1, \frac{2}{\theta} \right) \right] |\phi'(\varpi + \varrho - \xi)|^q \right\}.$$

Furthermore, we can get

$$\int_0^1 \left| \frac{1}{\theta^\eta} - \left( \frac{1 - (1-\varsigma)^\theta}{\theta} \right)^\eta \right| d\varsigma = \frac{1}{\theta^\eta} \left[ 1 - \frac{1}{\theta} B\left( \eta + 1, \frac{1}{\theta} \right) \right]. \tag{32}$$

By inserting (30)–(32) in (29), we gain the required inequality (28). □

**Remark 6.** *If we choose $\theta = 1$ in Theorem 5, then Theorem 5 reduces to [38] (Theorem 4).*

**Corollary 4.** *By the assumptions of Theorem 5 and assuming $\xi = \frac{\varpi + \varrho}{2}$, we get the next inequality*

$$\left| \frac{2^{\theta\eta - 1}\Gamma(\eta + 1)}{(\varrho - \varpi)^{\theta\eta}} \left[ {}^{\eta}Y_{\varrho}^{\theta}\phi\left(\frac{\varpi + \varrho}{2}\right) + {}_{\varpi}^{\eta}Y^{\theta}\phi\left(\frac{\varpi + \varrho}{2}\right) \right] - \phi\left(\frac{\varpi + \varrho}{2}\right) \right|$$

$$\leq \quad \frac{\varrho - \varpi}{4} \left( 1 - \frac{1}{\theta}B\left(\eta + 1, \frac{1}{\theta}\right) \right)^{1 - \frac{1}{q}}$$

$$\times \left[ \left[ \left( \frac{1}{4} - \frac{1}{\theta}B\left(\eta + 1, \frac{1}{\theta}\right) + \frac{1}{2\theta}B\left(\eta + 1, \frac{2}{\theta}\right) \right) |\phi'(\varrho)|^q \right. \right.$$

$$\left. + \left( \frac{1}{4} - \frac{1}{2\theta}B\left(\eta + 1, \frac{2}{\theta}\right) \right) |\phi'(\varpi)|^q \right]^{\frac{1}{q}} \tag{33}$$

$$+ \left[ \left( \frac{1}{4} - \frac{1}{\theta}B\left(\eta + 1, \frac{1}{\theta}\right) + \frac{1}{2\theta}B\left(\eta + 1, \frac{2}{\theta}\right) \right) |\phi'(\varpi)|^q \right.$$

$$\left. \left. + \left( \frac{1}{4} - \frac{1}{2\theta}B\left(\eta + 1, \frac{2}{\theta}\right) \right) |\phi'(\varrho)|^q \right]^{\frac{1}{q}} \right].$$

**Remark 7.** *If we choose $\theta = 1$ in Corollary 4, then Corollary 4 reduces to [38] (Corollary 2).*

**Theorem 6.** *Suppose $\phi$ is a function from $[\varpi, \varrho]$ into $\mathbb{R}$. If $\phi$ is differentiable on $(\varpi, \varrho)$ and $|\phi'|^q$ is concave for $\xi \in [\varpi, \varrho]$ and for some $q > 1$, then the next inequality holds for the fractional integrals ${}_{\varpi}^{\eta}Y^{\theta}$, ${}^{\eta}Y_{\varrho}^{\theta}$, and $\xi \in [\varpi, \varrho]$:*

$$\left| \frac{\theta^{\eta}\Gamma(\eta + 1)}{\varrho - \varpi} \left[ (\xi - \varpi)^{1 - \theta\eta}\, {}^{\eta}Y_{\varrho}^{\theta}\phi(\varpi + \varrho - \xi) + (\varpi - \xi)^{1 - \eta\theta}\, {}_{\varpi}^{\eta}Y^{\theta}\phi(\varpi + \varrho - \xi) \right] - \phi(\varpi + \varrho - \xi) \right|$$

$$\leq \quad \frac{\theta^{\eta}}{\varrho - \varpi} \left( \frac{1}{\theta^{\eta}} - \frac{1}{\theta^{\eta + 1}}B\left(p\eta + 1, \frac{1}{\theta}\right) \right)^{\frac{1}{p}} \tag{34}$$

$$\times \left( (\xi - \varpi)^2 \left| \phi'\left( \frac{a + 2\varpi - \xi}{2} \right) \right| + (\varpi - \xi)^2 \left| \phi'\left( \frac{a + 2\varpi - \xi}{2} \right) \right| \right),$$

*where $\frac{1}{q} + \frac{1}{p} = 1$.*

**Proof.** According to Hölder's inequality and Lemma 1, we have

$$\left| \frac{\theta^{\eta}\Gamma(\eta + 1)}{\varrho - \varpi} \left[ (\xi - \varpi)^{1 - \theta\eta}\, {}^{\eta}Y_{\varrho}^{\theta}\phi(\varpi + \varrho - \xi) + (\varpi - \xi)^{1 - \eta\theta}\, {}_{\varpi}^{\eta}Y^{\theta}\phi(\varpi + \varrho - \xi) \right] - \phi(\varpi + \varrho - \xi) \right|$$

$$\leq \quad \frac{\theta^{\eta}(\xi - \varpi)^2}{\varrho - \varpi} \left( \int_0^1 \left| \frac{1}{\theta^{\eta}} - \left( \frac{1 - (1 - \varsigma)^{\theta}}{\theta} \right)^{\eta} \right|^p d\varsigma \right)^{\frac{1}{p}} \left( \int_0^1 |\phi'(\varsigma\varrho + (1 - \varsigma)(\varpi + \varrho - \xi))|^q d\varsigma \right)^{\frac{1}{q}} \tag{35}$$

$$+ \frac{\theta^{\eta}(\varpi - \xi)^2}{\varrho - \varpi} \left( \int_0^1 \left| \frac{1}{\theta^{\eta}} - \left( \frac{1 - (1 - \varsigma)^{\theta}}{\theta} \right)^{\eta} \right|^p d\varsigma \right)^{\frac{1}{p}} \left( \int_0^1 |\phi'(\varsigma\varpi + (1 - \varsigma)(\varpi + \varrho - \xi))|^q d\varsigma \right)^{\frac{1}{q}}.$$

From the concavity of $|\phi'|^q$ and Jensen's integral inequality, we get

$$\int_0^1 |\phi'(\varsigma\varrho + (1-\varsigma)(\omega + \varrho - \xi))|^q d\varsigma$$

$$= \int_0^1 \varsigma^0 |\phi'(\varsigma\varrho + (1-\varsigma)(\omega + \varrho - \xi))|^q d\varsigma \tag{36}$$

$$\leq \left(\int_0^1 \varsigma^0 d\varsigma\right)\left|\phi'\left(\frac{1}{\int_0^1 \varsigma^0 d\varsigma}\int_0^1 \varsigma^0(\varsigma\varrho + (1-\varsigma)(\omega + \varrho - \xi))d\varsigma\right)\right|^q$$

$$= \left|\phi'\left(\frac{a + 2\omega - \xi}{2}\right)\right|^q.$$

Similarly,

$$\int_0^1 |\phi'(\varsigma\omega + (1-\varsigma)(\omega + \varrho - \xi))|^q d\varsigma \leq \left|\phi'\left(\frac{2\omega + \varrho - \xi}{2}\right)\right|^q. \tag{37}$$

Applying inequality (24) yields

$$\int_0^1 \left|\frac{1}{\theta\eta} - \left(\frac{1 - (1-\varsigma)^\theta}{\theta}\right)^\eta\right|^p d\varsigma \leq \frac{1}{\theta\eta}\int_0^1 \left(1 - \left(1 - (1-\varsigma)^\theta\right)^{p\eta}\right)d\varsigma \tag{38}$$

$$= \frac{1}{\theta\eta}\left(1 - \frac{1}{\theta}B\left(p\eta + 1, \frac{1}{\theta}\right)\right).$$

Hence, by considering inequalities (36)–(38) in (35), the desired inequality is obtained. $\quad\square$

**Remark 8.** *If we choose $\theta = 1$ in Theorem 6, then Theorem 6 reduces to [38] (Theorem 5).*

**Corollary 5.** *Consider the assumptions of Theorem 6. if we take $\xi = \frac{\omega + \varrho}{2}$, then we get the inequality*

$$\left|\frac{2^{\theta\eta - 1}\Gamma(\eta + 1)}{(\varrho - \omega)^{\theta\eta}}\left[{}^\eta Y_\varrho^\theta \phi\left(\frac{\omega + \varrho}{2}\right) + {}_\omega^\eta Y^\theta \phi\left(\frac{\omega + \varrho}{2}\right)\right] - \phi\left(\frac{\omega + \varrho}{2}\right)\right| \tag{39}$$

$$\leq \frac{\theta\eta(\varrho - \omega)}{4}\left(\frac{1}{\theta\eta} - \frac{1}{\theta\eta + 1}B\left(p\eta + 1, \frac{1}{\theta}\right)\right)^{\frac{1}{p}}\left[\left|\phi'\left(\frac{\omega + 3\varrho}{4}\right)\right| + \left|\phi'\left(\frac{3\omega + \varrho}{4}\right)\right|\right].$$

**Remark 9.** *If we choose $\theta = 1$ in Corollary 5, then Corollary 5 reduces to [38] (Corollary 3).*

### 3. Conclusions

In this paper, new midpoint type inequalities were investigated via the recently generalized Riemann–Liouville fractional integrals. An identity for a certain family of differentiable functions was proved in the framework of the suggested fractional integrals. Using this identity and the characteristics of convex and concave functions, several generalized midpoint type inequalities were proved. It was obvious that the results acquired in this paper could be reduced to the results of Budak and Agarwal in [38] when $\theta = 1$, and the results of Kirmaci [37] when $\theta = \eta = 1$.

**Author Contributions:** Methodology and conceptualization, A.-A.H., H.B., and A.A.A.; data curation and writing—original draft, A.-A.H., H.B. and A.A.A.; investigation and visualization, A.-A.H., H.B. and A.A.A.; validation, writing—reviewing, and editing, A.-A.H., H.B. and A.A.A. All authors have read and agreed to the published version of the manuscript.

**Funding:** This research was funded by King Khalid University, Grant RGP.2/15/43.

**Institutional Review Board Statement:** Not applicable.

**Informed Consent Statement:** Not applicable.

**Data Availability Statement:** The corresponding author will provide the data used in this work upon reasonable request.

**Acknowledgments:** The authors extend their appreciation to the Deanship of Scientific Research at King Khalid University for funding this work through Research Groups Program under grant RGP.2/15/43.

**Conflicts of Interest:** The authors declare no conflict of interest.

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
