# Peer review of "Further Midpoint Inequalities via Generalized Fractional Operators in Riemann–Liouville Sense"

_fractalfract, doi:10.3390/fractalfract6090496_

Round 1

Reviewer 1 Report

The authors study new midpoints inequalities related to some generalized fractional integrals.

In presenting the main results, the authors do not sufficiently highlight their relevance to the theory of fractional integral inequalities and do not sufficiently  describe their possible applications.
In fact, the body of the article appears as a succession of statements and proofs. And, what's more, many of the steps in the proofs consist of elementary calculations of mathematical analysis, so that the level of mathematics doesn't seem very high.

Surely, I have the impression that the results presented may be of some interest to some researchers. But, as it is written, the article cannot be accepted before a thorough revision and a profound improvement of the presentation.

Author Response

Point 1: The authors study new midpoint inequalities related to some generalized fractional integrals.

In presenting the main results, the authors do not sufficiently highlight their relevance to the theory of fractional integral inequalities and do not sufficiently describe their possible applications.

In fact, the body of the article appears as a succession of statements and proofs. And, what's more, many of the steps in the proofs consist of elementary calculations of mathematical analysis, so that the level of mathematics doesn't seem very high.

Surely, I have the impression that the results presented may be of some interest to some researchers. But, as it is written, the article cannot be accepted before a thorough revision and a profound improvement of the presentation.

Response 1: Thanks for valuable comments. Since our results generalize several published papers, we think that our paper deserves publication in Fractal and Fractional. In the revised version, we have done several changes to improve the paper. I hope you will reevaluate it as it is and consider it worthy of publication.

Reviewer 2 Report

 The paper represent some new results and is of interest.  So I think the paper is suitable for Fractal and fractional. But, the following changes are required before acceptance.

(1) Page2, in Definition 2, formula (5) is wrong, please check carefully.

(2) Page6, in Corollary 2, x should be .

Author Response

We are very grateful to the reviewer for his valuable comments, and we hope our responses to his comments are satisfactory. In the revised manuscript, the changes according to the reviewer’s comments are written in blue.

The paper represent some new results and is of interest.  So I think the paper is suitable for Fractal and fractional. But, the following changes are required before acceptance.

Point 1: Page2, in Definition 2, formula (5) is wrong, please check carefully.

Response 1: We have carefully checked formula (5) and fixed the error in it.

Point 2: Page6, in Corollary 2, x should be \xi .

Response 2: In Corollary 2, we have changed x to \xi .  

Reviewer 3 Report

This paper present several inequalities for a generalized fractional integral, the following are some comments:

1. The symbols used in the paper is not good to read, suggested use some common symbols.

2. In definition 2(generalized fractional Riemann-Liouville integrals), the right-fractional Riemann-Liouville integrals is not correct, please check. 

3. The describe of Definition 2  can be more detail, such as point out the order of fractional integral.

4. The relation between the general Riemann-Liouville integral and generalized Riemann-Liouville integral are suggest added.

5. There are some mistakes in Lemma 1.   

 (1) In the second line of equation(9), "\phi' "should be “\phi“”;

 (2) In the third line of equation(9), the first "-"should be "+";

 (3)In the fourth line of equation(9), the" (\varpi-\varsigma) "should be "(\varrho-\varsigma)"ï¼›

Please check Lemma 1 and make sure the result is correct.

6. In Lemma 3, the definition of Euler Beta function is suggested to be given.

7. In Corollary 2,  "If x=\frac{\varpi+\varrho}{2}", however, there is no "x" in Theorem 3,  so, what is the "x"?

8. In Figure 1, the "x-label" are suggested to be added.

9. The proof of the theorems should be written in more detail. For example, in equation(29), the first inequality is not obtained by power mean inequality directly, some auxiliary tools are suggested to be added or described.

10. What is the meaning of the generalized Riemann-Liouville integral? Is there any applications about it?

11. The importance of the generalized Riemann-Liouville integral and the inequalities are suggested to be described in Introduction part.

Author Response

We are very grateful to the reviewer for his valuable comments, and we hope our responses to his comments are satisfactory. In the revised manuscript, the changes according to the reviewer’s comments are written in red.

This paper presents several inequalities for a generalized fractional integral, the following are some comments:

Point 1: The symbols used in the paper is not good to read, suggested use some common symbols.

Response 1: We respect the suggestion of the reviewer for using common symbols. But using common symbols will increase the similarity index of our manuscript. To our knowledge, the used symbols are the known Greek symbols and can be read (for examples:  $\varpi$ is varpi, $\varrho$ is varrho, and $\varsigma$  is varsigma). Also, they are usually used in several published works. Moreover, we propose these symbols to avoid Plagiarism in our manuscript.

Point 2: In definition 2 (generalized fractional Riemann-Liouville integrals), the right-fractional Riemann-Liouville integrals is not correct, please check.

Response 2: We have checked the right-fractional Riemann-Liouville integrals in definition 2 and fixed the typo in it. In the revised version, the changes are in blue. 

Point 3: The describe of Definition 2 can be more detail, such as point out the order of fractional integral. 

Response 3: We have described Definition 2 in more detail and pointed out the order of fractional integral.

Point 4: The relation between the general Riemann-Liouville integral and generalized Riemann-Liouville integral are suggest added.

Response 4: We have added Remark 1 to mention the relation between the general Riemann-Liouville integral and the generalized Riemann-Liouville integral.

Point 5: There are some mistakes in Lemma 1.

  • In the second line of equation(9), "\phi' "should be “\phi“”;

  • In the third line of equation(9), the first "-"should be "+";

  • In the fourth line of equation(9), the" (\varpi-\varsigma) "should be "(\varrho-\varsigma)"ï¼›

Please check Lemma 1 and make sure the result is correct.

Response 5: We have checked Lemma 1, made sure the result is correct, and corrected all the above typos ((1), (2), and (3)) mentioned by the reviewer.

Point 6: In Lemma 3, the definition of Euler Beta function is suggested to be given.

Response 6: In Theorem 3, we have added the definition of Euler Beta function.

Point 7: In Corollary 2,  "If x=\frac{\varpi+\varrho}{2}", however, there is no "x" in Theorem 3,  so, what is the "x"?

Response 7: This is a typo. In Corollary 2, we have changed “x” to “ \xi”. the change is in blue.

Point 8: In Figure 1, the "x-label" are suggested to be added.

Response 8: In Figure 1, we have added the "x-label".  

Point 9: The proof of the theorems should be written in more detail. For example, in equation(29), the first inequality is not obtained by power mean inequality directly, some auxiliary tools are suggested to be added or described.

Response 9: If we apply the power mean inequality in the inequality (12), then we can obtain the inequality (29). We have corrected it.

Point 10: What is the meaning of the generalized Riemann-Liouville integral? Is there any applications about it?

Response 10: The generalized Riemann-Liouville integral and corresponding derivatives have many applications in differential equations and integral equations. One can see them by the citation of [18]. In this paper, we apply these integrals to generalize the classical midpoint type inequalities.

Point 11: The importance of the generalized Riemann-Liouville integral and the inequalities are suggested to be described in Introduction part.

Response 11: The most importance of these integrals is that they generalize the Riemann-Liouville fractional integrals by a parameter. One can see the relations between these integrals and Riemann-Liouville fractional integrals in newly added Remark 1. 

Reviewer 4 Report

In this article, the authors obtained new midpoint type inequalities are given through recently generalized Riemann-Liouville fractional integrals. Foremost, authors present an identity for a class of differentiable functions including the proposed fractional integrals. Thus, several midpoint type inequalities containing generalized Riemann-Liouville fractional integrals are proved by employing the features of convex and concave functions. The results herein, to the best of my knowledge, look interesting. I recommend this paper for a minor revision due to some following reasons

(1) The title is not suitable, because the title indicates different generalized types of fractional integral So change the article title.

       (2)    Add the paper construction paragraph in the last of introduction section.

(3)      Give comma and full stop at proper place throughout the paper.

   (4)    No one can ignore the importance of integral inequalities in the direction of pure and applied sciences. Integral inequalities of this form and the arrangement with special cases performs a good contribution.” Can the authors support this with the help of example? 

     (5)    I found that there are many recently published literatures on different type of fractional integral and differential operator, I suggest the author to add about inequalities obtained by using Caputo–Fabrizio, Marichev-Saigo-Maeda and Generalized Proportional Hadamard Fractional Integral Operators, which will be improve the introduction of this paper.

Author Response

We are very grateful to the reviewer for his valuable comments, and we hope our responses to his comments are satisfactory. In the revised manuscript, the changes according to the reviewer’s comments are written in green.

In this article, the authors obtained new midpoint type inequalities are given through recently generalized Riemann-Liouville fractional integrals. Foremost, authors present an identity for a class of differentiable functions including the proposed fractional integrals. Thus, several midpoint type inequalities containing generalized Riemann-Liouville fractional integrals are proved by employing the features of convex and concave functions. The results herein, to the best of my knowledge, look interesting. I recommend this paper for a minor revision due to some following reasons.

Point 1: The title is not suitable, because the title indicates different generalized types of fractional integral, so change the article title.

Response 1: In the title of the revised version, we have specified the used generalized fractional integral and changed the title from “Further Midpoint Inequalities via Some Generalized Types of Fractional Operators” into “Further Midpoint Inequalities via Generalized Fractional Operators in Riemann-Liouville Sense “.

Point 2: Add the paper construction paragraph in the last of introduction section.

Response 2: We have added the paper construction paragraph in the last of introduction section.  

Point 3: Give comma and full stop at proper place throughout the paper.

Response 3: In all parts of our manuscript, we have fixed the punctuations by inserting comma or full stop at proper place.

Point 4:  No one can ignore the importance of integral inequalities in the direction of pure and applied sciences. Integral inequalities of this form and the arrangement with special cases performs a good contribution.” Can the authors support this with the help of example?

Response 4: In fact, we have an example that supports our results. It did not appear due to a code error. Now you can see it after Remark 3.

Point 5: I found that there are many recently published literatures on different type of fractional integral and differential operator, I suggest the author to add about inequalities obtained by using Caputo–Fabrizio, Marichev-Saigo-Maeda and Generalized Proportional Hadamard Fractional Integral Operators, which will be improve the introduction of this paper.

Response 5: Thanks for your valuable suggestions. We have added some papers on Caputo–Fabrizio, Marichev-Saigo-Maeda and Generalized Proportional Hadamard Fractional Integral Operators in the references list. ([31-33]).

Round 2

Reviewer 1 Report

Accepted in present form.

Reviewer 3 Report

the authors modified the paper according to the reviewer's advise, I suggest the paper be accepted.